# Genome-Wide Association Studies of Salt Tolerance at the Seed Germination Stage and Yield-Related Traits in *Brassica napus* L.

**DOI:** 10.3390/ijms232415892

**Published:** 2022-12-14

**Authors:** Yan Zhang, Ping Li, Jie Zhang, Yaqi Li, Aixia Xu, Zhen Huang

**Affiliations:** State Key Laboratory of Crop Stress Biology for Arid Areas, College of Agronomy, Northwest A&F University, Yangling, Xianyang 712100, China

**Keywords:** *Brassica napus*, GWAS, SLAF-seq, salt stress, yield-related traits, candidate genes

## Abstract

Salt stress severely affects crop growth and development and reduces the yield of *Brassica napus*. Exploring natural genetic variations for high salt tolerance in *B. napus* seedlings is an effective approach to improve productivity under salt stress. Using 10,658 high-quality single nucleotide polymorphic (SNP) markers developed by specific-locus amplified fragment sequencing (SLAF-seq) technology, genome-wide association studies (GWAS) were performed to investigate the genetic basis of salt tolerance and yield-related traits of *B. napus*. The results revealed that 77 and 497 SNPs were significantly associated with salt tolerance and yield-related traits, of which 40 and 58 SNPs were located in previously reported QTLs/SNPs, respectively. We identified nineteen candidate genes orthologous with *Arabidopsis* genes known to be associated with salt tolerance and seven potential candidates controlling both salt tolerance and yield. Our study provides a novel genetic resource for the breeding of high-yield cultivars resistant to salt stress.

## 1. Introduction

Rapeseed (*Brassica napus* L.) is the second largest oilseed crop in the world, which provides vegetable oil for humans as well as being used in a variety of other ways, such as being used as vegetables, bio diesel fuel, animal feed, and ornamental flowers [1]. According to the United States Department of Agriculture (USDA) forecast data, global rapeseed oil production declined by 5.86% in the years 2021/2022. In China, the self-sufficiency rate of vegetable oil is about 40%, which is far below the safety level of 60% [2]. With the rapidly growing demand for rapeseed oil, increasing the yield of oilseed crops is always a major breeding objective. Plants under field conditions are subjected to various environmental stresses, and salt stress has emerged as a major factor. It is estimated that approximately 20% of the total cultivable lands and 33% of the irrigated agricultural lands are affected by salinity in the world, and 15% of the irrigated lands are affected by salinity in China [3], which has an important impact on agricultural production. Moreover, rapeseed planting areas have significantly decreased, and salt stress significantly restricts the growth and production of rapeseed [4]. Therefore, there is an urgent need to improve the environmental adaptation of *B. napus* so as to utilize the saline-alkaline land, increase the cultivation area and improve the yield. Unraveling the genetic and molecular mechanisms of salt tolerance is a critical step for improving yield.

Salt stress affects the growth of *B. napus* at different stages of the plant’s growth [5,6], among which the germination and early seedling stages are more sensitive to salinity than the other growth stages, and consequently these effects lead to yield and biomass reduction [7]. The salt tolerance traits and yield-associated traits are complex quantitative traits that are controlled by numerous loci in *B. napus* [8,9]. Quantitative trait locus (QTL) mapping is one of the most common approaches for the genetic analysis of quantitative traits. Many studies have been conducted to identify QTLs for seed yield and yield-related traits in *B. napus* based on traditional QTL mapping [10,11,12]. In contrast, QTLs associated with adaptation to salt stress are comparatively rare [13]. However, the resolution of these studies is low, time-consuming and costly due to the need to create large-scale segregation populations and high-density genetic maps. With the development of next-generation sequencing (NGS) technology, the genome-wide association study (GWAS) has become an effective approach to map QTLs related to complex traits of crops and has been applied in various crops, such as rice, soybean, and *B. napus* [14,15,16]. Some QTLs and candidate genes associated with salt-tolerance-related traits at the germination and seedling stages [17,18,19,20,21,22,23] and yield-related traits [24,25,26] have gradually been identified through GWAS in *B. napus*. Although a number of QTLs and candidate genes have been identified for salt tolerance, the molecular mechanisms underlying the regulation of salt tolerance and yield remain unclear in *B. napus.*

It is an important step to detect single nucleotide polymorphism (SNP) markers in GWAS analysis. Recently, specific-locus amplified fragment sequencing (SLAF-seq), as a fast, accurate, highly efficient and cost-effective method, has been applied to generate a huge number of SNPs for GWAS studies. This approach has been used in many crops, such as rice [27], soybean [28], tea [29], citrus [30], and rapeseed [31]. However, no previous studies have been reported to use the SLAF-seq technology of GWAS analysis for salt-tolerance traits at the germination stage in *B. napus.* In this study, we used 10,658 SNP markers developed from SLAF-seq to genotype a panel of 146 rapeseed accessions. GWAS was performed with the combination of three salt-tolerance traits and six yield-related traits, and SNP markers. The present study aimed to identify salt tolerance and yield-related SNPs and candidate genes in rapeseed, which will help us better understand the genetic basis of *B. napus* breeding for yield and salt tolerance traits.

## 2. Results

### 2.1. Phenotypic Variation

The descriptive statistics of all traits are summarized in Table 1. All traits showed significant variation in all environments, and the coefficient of variance (CV) ranged from 13.73% to 133.00% (Table 1, Appendix A). Compared with other yield-related traits, silique length (SL) and seeds per silique (SS) showed the lowest coefficient of variance (CV), while the seed yield per plant (SYP) exhibited the highest CV (Table 1). Interestingly, high variability for germination vigor (GV), germination rate (GR), and relative salt damage index (RSDI) traits was observed, showing 133%, 120%, and 20%, respectively. This suggests that our materials had a strong heterogeneity for salt-related traits. Among them, two germplasms had a strong tolerance to salt stress with a GV and GR of more than 70%, and RSDI of less than 30%. Overall, the GV and GR of more than 76% accessions ranged from 0 to 20%, and the RSDI of 77% accessions ranged from 80 to 100%.

The Pearson correlation coefficients among all traits were analyzed, and strong correlations were observed. For example, RSDI was significantly negatively correlated with GV and GR (Figure 1). SYP showed significant positive correlations with all yield-associated traits across different years except SL, especially DWP with a correlation coefficient of 0.70. SMI was significantly negatively correlated with SL and TSW, and significantly positively correlated with SYP and DWP. Interestingly, TSW was significantly negatively correlated with GV and GR, and positively correlated with RSDI. Taken together, these results demonstrate that there is a complex association between seed yield and other traits, either directly or indirectly, including salt tolerance traits and yield-related traits.

### 2.2. Genotyping Analysis and Linkage Disequilibrium

A total of 336.22 Mb paired-end reads were obtained with a cutoff efficiency of 90.97%. The average percentage of Q30 bases was 92.30% and the GC content was 40.57%, indicating that the sequencing results for 146 *B. napus* accessions were reliable. A total of 665,207 SLAF tags were detected, including 304,506 polymorphic SLAF tags. The average sequence depth of each SLAF tag was 12.30. These SLAF tags were evenly distributed on each chromosome (Appendix A; Appendix A). SLAF tags on chromosome A10 had the highest polymorphic rate (57.52%), while the SLAF tags on chromosome C09 had the lowest polymorphic rate (33.78%). Furthermore, a total of 1,620,818 SNPs were developed from 304,506 polymorphic SLAF tags (Appendix A), and 10,658 SNPs were selected as the candidate SNPs for GWAS analysis using the criteria of integrity ratio of >0.8 and MAF of >0.05. LD was estimated as the R^2^ value. It was found that R^2^ ranged from 0.26 on chromosome A05 to 0.37 on chromosome C01, with an average of 0.3, revealing the differences in the level of LD among chromosomes (Appendix A). The average LD decay distance was about 300.33 kb at a threshold of R^2^ = 0.1 (Figure 2).

### 2.3. Population Structure and Kinship Analyses

The population structure was analyzed using Admixture software, and the cross-validation (CV) error rate for K = 1–10 was examined. As shown in Figure 3A,B, when K = 3, the CV dropped to the lowest value, which indicated that the entire population could be divided into three subgroups. The phylogenetic tree analysis also showed that the 146 accessions could be assigned to three groups (Figure 3C). Furthermore, the genetic variance between accessions was conducted by PCA analysis. The first three components explained 7.02, 5.38, and 2.48% of the variance, respectively (Figure 3D). The kinship was evaluated using SPAGEDi software (Figure 3E). The results showed that 85.2% accessions had a genetic similarity coefficient lower than 0.05, indicating that the genetic relatedness among rapeseed individuals was low.

### 2.4. Genome-Wide Association Analysis for Salt-Related Traits

To reduce the influence of population structure and improve the reliability of GWAS results, we used five statistical models, GLM, MLM, FaST-LMM, EMMAX, and CMLM. Model fit was tested for each trait, and the most appropriate statistical model was selected based on no inflation of most of the *p* values with a sharp upward deviated tail in the Q-Q plots. The results showed that the FaST-LMM and GLM models were the most appropriate models for the three salt-related traits (Figure 4A). A total of 77 significant SNPs associated with three salt tolerance traits were detected using FaST-LMM and GLM analyses distributed on all chromosomes except A08 (Figure 4B; Appendix A). Among them, 53 SNPs were obtained by FaST-LMM analysis, 28 by GLM analysis, and 4 by both FaST-LMM and GLM analyses (Appendix A). For germination vigor (GV), 54 SNPs were identified; 33 of these were detected by FaST-LMM and 3 by both FaST-LMM and GLM models. Among them, 24 SNPs were identified simultaneously in two replicate experiments. For the germination rate (GR), 36 SNPs were identified; 28 SNPs were detected by FaST-LMM and 4 SNPs by both FaST-LMM and GLM analyses. Among them, 20 SNPs were identified simultaneously in two replicate experiments. Similarly, 32 SNPs were detected by relative salt damage index (RSDI), 24 SNPs by FaST-LMM and 4 SNPs by both FaST-LMM and GLM analyses. Among them, 15 SNPs were identified simultaneously in two replicate experiments. A total of 36 significant SNPs associated with three salt tolerance traits were detected simultaneously in two replicate experiments (Appendix A). Furthermore, 30 SNPs were associated with more than one trait, and explained individually from 11.20 to 17.68% of the phenotypic variance, including 10 SNPs detected in one replicate experiment and 20 SNPs detected simultaneously in two replicate experiments (Appendix A).

### 2.5. Genome-Wide Association Analysis for Yield-Related Traits

Model fit was tested for six yield-related traits based on Q-Q plots, and the results showed that GLM was the appropriate model for TSW and SYP, FaST-LMM was the appropriate model for SS, and FaST-LMM and GLM were the appropriate models for SL, SMI and DWP (Figure 5). A total of 497 significant SNPs associated with six yield-related traits were detected, which were distributed on all chromosomes and explained individually from 6.89 to 31.83% of the phenotypic variance (Figure 6; Appendix A; Appendix A). These SNPs were combined to form 237 QTLs, including 78 for SL, 38 for SS, 59 for SMI, 24 for TSW, 10 for SYP, and 28 for DWP. Most of the QTLs were only detected in one year and 27 QTLs composed of 13 intervals were repeatedly detected in at least two years. These repeatable QTLs included 11 for SL distributed on A02, A03, A04, A09 and C04, 12 for SMI distributed on A05, C02, C03 and C08, 2 for SS on A06, and 2 for DWP on A02 (Appendix A). Notably, we found some pleiotropic loci controlling two to three different traits, which were named as pleiotropic QTLs (*pq*QTLs) accordingly. A total of 69 QTLs were integrated into 31 *pq*QTLs, of which two pqQTLs (*pqA2.2* and *pqA4.1*) had the same regions (Appendix A). Other *pq*QTLs, *pqA2.1* for SYP and DWP, *pqA3.2* for SL and SYP, *pqA9.1* for SL and SMI, *pqA9.3* for SL and TSW were identified, and contained 11, 9, 18, and 12 significantly associated SNPs, respectively. Moreover, four QTLs controlling three different traits were identified, including *pqA8.2* for SS, TSW, and SMI, *pqA9.2* for SL, SYP, and SMI, *pqC2.1* for SMI, TSW, and SL, and *pqC3.2* for SL, SMI, and SS.

### 2.6. Associated SNPs for Salt Tolerance Co-Localized with QTLs for Yield-Related Traits

To explore the connection between salt tolerance traits and yield-related traits, salt-associated SNPs identified in the current study were compared with QTLs for yield-related traits. A total of 18 salt-associated SNPs were co-located with 8 QTLs for yield-associated traits, including 2 pleiotropic QTLs (*pqA2.1* and *pqC9.1*) and 3 repeatable QTLs (*qSL18–A2.1*, *qSL19–A2.2*, and *qSMI20–A5.1*). Seven of the eighteen salt-associated SNPs were pleiotropic loci and were detected simultaneously in two replicate experiments. Notably, the QTL *pqC9.1* for DWP and SS contained six strongly associated SNPs, among which two SNPs (C09: 12,521,248 bp, and 12,521,335 bp) for DWP overlapped with rsC09_12521248 and rsC09_12521335 for GR and GV/GR, respectively (Table 2).

### 2.7. Comparative Genome Analysis for Salt Tolerance and Yield-Related Traits of the Present and Previous QTLs

A total of 40 SNPs may overlap with salt-related SNPs identified previously, among which 16 SNPs were pleiotropic loci and were detected simultaneously in two replicate experiments (Appendix A). Four SNPs (ChrA02: 8,228,124 bp; ChrA03: 16,375,764 bp; ChrA06: 22,471,588, and 22,471,644 bp) for GV and GR were co-localized with the QTLs identified previously. Notably, the three SNPs (ChrA10: 15,189,187, 15,189,200, and 15,189,284 bp) for SII and the SNP (ChrA10, site: 15,188,859 bp) for multiple traits (GV/GR/RSDI) were repeatedly detected by previous studies for plant height (PH), ground dry weight (GDW), and yield (SYP) at the mature stage. Furthermore, the four SNPs were co-localized with the QTL *qSFWs-A10* (15,181,794–17,289,298 bp) on chromosome A10 in our previous study (unpublished). In addition, 26 SNPs and 7 SNPs were less than 100 kb and 300 kb away from 84 and 28 salt-related SNPs identified previously, respectively. For yield-related traits, a total of 28 QTLs containing 58 SNPs overlapped with QTLs identified previously, including 11 QTLs for SL, 6 QTLs for SS, 2 QTLs for SMI, 6 QTLs for TSW, 2 QTLs for SYP, and 1 QTL for DWP (Appendix A). Among the 28 QTLs, 12 QTLs were pleiotropic QTLs. Interestingly, a pleiotropic QTL *pqA9.3* for SL and TSW (the present study) was identical to seven QTLs for SL in four different experiments, and five QTLs for TSW in three different experiments (previous study).

### 2.8. Candidate Genes for Controlling Both Salt-Alkali Tolerance and Yield

As shown in Appendix A, a total of 36 significant SNPs associated with three salt tolerance traits were detected simultaneously in two replicate experiments, of which 8 SNPs were co-localized SNPs for salt tolerance and yield-related traits (Table 2). Significant SNPs were mapped to the ‘Darmor-*bzh*’ reference genome (*Brassica napus* v.4.1), and the candidate genes were selected within 100 kb flanking sequences of a significant SNP. A total of 732 genes were associated with the 36 significant SNPs (Appendix A). In order to identify the salt-stress-related candidate genes of rapeseed, we referred to the genome annotation information of *Arabidopsis*. Among these 732 genes, 514 were homologous to 477 *Arabidopsis* genes. Of these, 19 genes were putatively related to salt tolerance based on GO annotation, such as response to osmotic, water, water deprivation, and salt stresses (Table 3), of which 15 genes were located in the A genome, and 4 genes in the C genome. The physical distance between these candidate genes and the significant SNPs varied from 1.5 to 99.3 kb (Table 3). Among these genes, 18 genes were identified for GV, 17 genes were identified for GR, and 13 genes were identified for RSDI. Notably, 16 genes were associated with more than one trait. The 19 candidate genes belonged to various groups, such as transcription factors, kinases, phosphatases, calcium-binding proteins, and transporters, and are closely associated with 18 SNPs. Seven of these eighteen SNPs, including *BnaA02g14490D*, *BnaA02g05520D*, *BnaA02g05590D*, *BnaA05g11750D*, *BnaA05g11880D*, *BnaC09g15950D*, and *BnaC09g16050D*, were co-located with QTLs for yield-associated traits, and detected simultaneously in two replicate experiments. These seven genes are potential candidates controlling both salt tolerance and yield.

## 3. Discussion

Good seedling establishment is critical to crop yield, and one of the key factors is the seed germination performance, which is directly affected by seed dormancy and vigor. Germination speed is a major key element of vigorous seeds [32,33]. Previous investigations have shown that seedlings with better germination rates under non-stress conditions also have advantages under stress conditions [34]. In this study, we determined the response of 146 *B. napus* accessions to salt stress at the seed germination stage. Three phenotypic traits were measured at the seed germination stage. Large phenotypic variations for the three traits were observed in these 146 *B. napus* accessions, suggesting GV, GR, and RSDI are good indexes for our study (Table 1). We found that more than 76% accessions decreased in GV and GR, and increased in RSDI, indicating the importance of the seed germination stage to plant growth. Different genotypes of plants have different tolerances to salt stress [35]. Here, two accessions simultaneously exhibited higher GV and GR, and lower RSDI under salt stress, which is consistent with the strong correlation between GV, GR, and RSDI. These specific materials may contain key salt-tolerant genes and will be good sources for molecular breeding of salt tolerance. SYP was significantly correlated with all traits except for SS, but a significant correlation was observed between SYP and SS in a previous study [36]. In addition, no significant correlation was observed between TSW and SS, which is not consistent with a previous study [12]. This indicates that agronomic traits are complex quantitative traits and are diversified under different genetic backgrounds.

We identified a total of 665,207 polymorphic SLAF tags containing 1,620,818 SNPs_s_, and finally selected 10,658 high consistency SNPs. The polymorphism rate was slightly higher for the A subgenome than for the C subgenome in our population, indicating that the genetic diversity of the A subgenome was higher than the C subgenome (Appendix A). This result is consistent with the results of Li et al. [24], while it is contrary to the results of Delourme et al. [37]. In our study and the study of Li et al., most of the genotypes were Chinese rapeseed cultivars, while those of Delourme et al. were European. Previous studies have shown that there is extensive genetic diversity in three ecotypes (semi-winter, winter, and spring) of *B. napus* [38,39]. Our study showed that the entire population could be divided into three subgroups although all the germplasms were winter types. This may be due to ecogeographic adaptation and human selection. The kinship analysis also showed only a weak or absent relationship among most materials, together with significant variance in phenotype performance, indicating that this population is suitable for association analysis. Population structure strongly influences LD patterns [40]. Here, we found LD decayed dramatically faster in the A subgenome than in the C subgenome (Appendix A), which was consistent with previous studies [41]. Higher genetic diversity and LD decay in the A subgenome indicated that the A subgenome recombination rate was higher than the C subgenome. The primary reason for this is thought to be that Chinese rapeseed cultivars have been selected for improved adaptation to local environments by introgression of Chinese *B. rapa* into *B. napus* [38,42].

In order to improve the accuracy of GWAS results, we used five statistical models: GLM, MLM, FaST-LMM, EMMAX, and CMLM models. It has been previously suggested that multiple algorithmic models should be used for GWAS analysis of complex traits [31,43,44]. In our study, FaST-LMM and GLM models were the most effective models for all traits compared with other models. For three salt-related traits, a total of 77 SNPs were obtained by FaST-LMM and GLM models, while only 28 loci were detected by the GLM model (Appendix A). For each salt-related trait, the FaST-LMM model detected the maximum number of SNPs compared with the other models, in accordance with the previous studies [15]. However, for six yield-related traits, the GLM model fits all traits except for SS.

In our study, we identified 54 SNPs for germination vigor (GV), 36 SNPs for germination rate (GR), and 32 SNPs for relative salt damage index (RSDI). Among these, 30 SNPs were associated with multiple traits in accordance with the high correlation among the three traits (Appendix A). Additionally, 20 of 30 SNPs were detected simultaneously in two replicate experiments, indicating the reliability of these loci. Previous studies have explored the genetic basis of salt stress responses at the germination or seedling stage using traditional QTL mapping or GWAS analysis [13,17,18,19,20,21,22,23,45]. Compared with the previous results, four SNPs (ChrA10: 15, 189, 187, 15, 189, 200, 15, 189, 284, 15, 188, 859 bp) could be repeatedly detected at the seedling and mature stages [23], and controlled different traits, suggesting that the four SNPs are common loci that control the salt tolerance of rapeseed at different stages. In addition, 33 SNPs did not overlap with salt-related SNPs identified previously, but they were less than 300kb apart. Among these 33 SNPs, 5 SNPs on A02 (2, 496, 879, 8, 228, 124 bp) and A05 (6, 670, 181, 6, 692, 257, 6, 692, 275 bp) were not only detected simultaneously in two replicate experiments, but co-located with 4 QTLs for yield-associated traits, including 1 pleiotropic QTL *pqA2.1*, 2 repeatable QTLs (*qSL18–A2.1* and *qSL19–A2.2*), and 1 QTL *qSL18–A5.2*. Moreover, four of the five SNPs controlled more than one trait. These results indicate that these SNPs detected in the present study are reliable. Except for the above SNPs reported previously, all remaining 37 SNPs identified in our study should be considered novel. Interestingly, most of the novel SNPs are located on chromosome C, of which 11 SNPs were associated with more than one trait.

Contrary to the salt tolerance traits, QTLs or SNPs of yield-related traits have been frequently reported in *B. napus*. In this study, 497 SNPs/237 QTLs were obtained for yield-related traits under the three-year growth environment. Of these, 27 QTLs (repeatable QTLs) were repeatedly detected in two or three years (Appendix A), suggesting that the QTLs associated with these SNPs were stable under the growing environments [46]. The other QTLs were environment-specific, indicating that phenotypic plasticity plays an essential role in plant agronomic diversity [47]. A total of 28 QTLs overlapped with QTLs identified previously (Appendix A), of which a QTL for SL on A09 (27,745,423–28,839,525 bp) was repeatedly identified seven times in four different experiments. The QTL region was not only repeatedly identified at three years, but overlapped with the QTL *qTSW19–A9.1* for TSW. Interestingly, the QTL *qTSW19-A9.1* (28,050,920–28,051,279 bp) was also repeatedly identified four times in three different experiments. These results indicated that the QTL region for SL and TSW on A09 was stable in different genetic backgrounds and exhibited pleiotropic effects, which is consistent with the positive correlation between TSW and SL. These findings may be useful in molecular marker-assisted selection breeding.

Pleiotropism is a common phenomenon that has been found in the GWAS of rapeseed [48]. In our study, we identified 31 pleiotropic QTLs (*pq*QTLs) for different yield related traits (Appendix A). Of these, SNP-rsA04_4700698 exhibited a pleiotropic effect on TSW and SL. The SNP-rsA04_4700698-associated candidate gene was *BnaA04g06140D* which is homologous to *AT4G14240* (*CST2*) and encodes a CBS domain-containing protein with a domain of unknown function DUF21 of *Arabidopsis*. Knockout of *CST2* results in stomatal closing, decreased photosynthesis, and growth retardation [49]. In addition, five SNPs (rsA02_7701568, rsA02_7701573, rsA02_7701581, rsA02_7701637, and rsA02_7701673) exhibited a pleiotropic effect on SYP and DWP, which is consistent with the strong correlation between them. The five-SNP-associated candidate gene was *BnaA02g13910D* which is homologous to *AT1G68500*, which may function in the fatty acid metabolic process, organic substance catabolic process, response to cold, response to lipids, and response to water deprivation based on GO annotation. Moreover, we identified some pleiotropic loci between yield-related traits and salt tolerance traits (Table 2). Of these, SNPs rsC09_12521248 and rsC09_12521335 exhibited a pleiotropic effect on GV, GR, and DWP. The two-SNP-associated candidate gene was *BnaC09g15650D* which is homologous to *AT1G57680* (*CAND1*) and encodes a candidate G-protein-coupled receptor, which may function in cell growth, developmental growth involved in morphogenesis, the hormone-mediated signaling pathway, and regulation of developmental processes. Pollen germination and pollen tube growth is an essential process for the reproduction of flowering plants. The expression of *CAND1* is significantly changed during pollen germination and tube growth [50]. Another two G-protein-coupled receptors, *Cand2* and *Cand7*, are involved in the regulation of root growth in *Arabidopsis* [51]. Root development of plants is related to the environment. Plants can adapt to environmental stresses by modulating root growth, and water and nutrition extraction [52]. *CAND1* may have a similar function to *Cand2* and *Cand7*, which may be a candidate gene to regulate both yield and resistance for abiotic stresses. These pleiotropic SNPs will be helpful in understanding the molecular mechanism of rapeseed yield formation.

Currently, little is known about the salt tolerance and yield genes of *B. napus*. In this study, we identified 19 candidate genes linked with 18 significant SNPs for salt tolerance traits, of which 7 genes may regulate both salt resistance and yield of plants. Three genes (*OST2*, *NHD1* and *CAMTA1*) are related to osmotic adjustment and Na^+^ ion concentration. OST2 (OPEN STOMATA 2) encodes a plasma membrane H^+^ATPase, and the *ost2* mutation impairs stomatal response to ABA of *Arabidopsis* during its drought response [53]. The elimination of excess sodium ions in cells is a major mechanism for plants to deal with salt stress. The Na +/H + transporters play an important role in dealing with ion homeostasis under salinity [54]. NHD1, one of the isoforms of sodium-hydrogen-antiporter (NHAD)-type carriers, mediates sodium/proton antiport. The *nhd1* knockout mutants significantly decreased capacity for sodium export out and salt tolerance of *Arabidopsis* [55]. Plant CAMTAs are known to mediate responses to biotic and abiotic stresses [56]. The *camta1* mutant showed drought sensitivity, and CAMTA1 regulates “drought recovery” as the most indicative pathway along with ABA response, osmotic balance, DNA methylation, and photosynthesis [57].

It is well known that transcription factors (TFs) play an important role in response to salt stress in plants [58]. However, little is known about the role of TFs in salt stress responses in rapeseed. Two TFs, *ANAC55* (*BnaA03g33890D*), an ATAF-like NAC-domain transcription factor, and *MYB88* (*BnaA06g33990D*), a member of the R2R3 factor gene family, were identified in our candidate genes. These classes of transcription factors have also been predicted in previous studies to respond to salt or drought stresses [19,31]. The R2R3-type of the MYB gene is a large family within the MYB family. Some R2R3-MYB proteins have been found to be involved in responses to abiotic stresses [59]. Overexpression of *AtMYB44* enhances ABA-mediated stomatal closure to confer abiotic stress tolerance in transgenic *Arabidopsis* [60]. *ANAC55* is induced by drought, high salinity and heat stresses, and overexpression of this gene increases drought tolerance [61], while mutation enhances heat tolerance in *Arabidopsis* [62]. Additionally, *DRIP2*, *LEW2*, *M3KDELTA6*, *SNRK2.1*, *PEN2*, *STRS2*, and *OST2* belong to the enzyme functional group. Of these, *DRIP2* and *LEW2* are located in the region controlling both salt tolerance and yield-related traits. DRIP2 acts as a negative regulator in drought-responsive gene expression by targeting DREB2A to 26S proteasome proteolysis, and plant development, and the growth of the *drip1 drip2* double mutant were significantly delayed and prolonged compared with the wild-type and the two single mutants [63]. The mutants *lew2-1* and *lew2-2* show weaker or no wilting phenotype under normal conditions, but are more tolerant to salt and drought stresses [64]. *M3KDELTA6* and *SNRK2.1* encode two serine/threonine-protein kinases. There are ten SnRK2 kinases in *Arabidopsis*, and at least nine of these are activated in response to osmotic stresses [65]. The MAP kinase kinase kinases (M3Ks) family is essential for reactivation of SnRK2 protein kinases after PP2C dephosphorylation [66]. *STRS2* encodes an RNA helicase protein and is identified as an upstream negative regulator of *Arabidopsis* in response to multiple abiotic stresses [67]. Besides these genes, five other genes (*SIED1*, *VAMP714*, *HVA22C*, *PGN*, and *CER1*) related to salt tolerance are also located in the region controlling both salt tolerance and yield-related traits, of which *BnHVA22C* (*BnaA02g14490D*) has been mapped in previous studies [23,45]. Of the five genes, two genes (*SIED1* and *CER1*) have been reported to be associated with plant growth and development. The mutant *cer1-1* reduces the production of stem epicuticular wax, pollen fertility under conditions of low humidity, and reduced plant height. The overexpression of *CsCER1* alters cuticular wax biosynthesis and enhances drought resistance of cucumber [68]. Using map-based cloning, a *Ms-cd1* gene, a homolog of the *Arabidopsis SIED1* gene, has been identified, which is responsible for the male sterile phenotype in *Brassica oleracea*, and the gene exhibits higher expression in the male sterile lines but not in the fertile lines [69]. Taken together, among the seven genes located in salt tolerance and agronomy co-localization, four genes have been reported to be related to the growth and stress tolerance of plants. However, three genes (*DRIP2*, *SIED1* and *LEW2*) showed opposite phenotypes between the development and resistance of plants, and only one gene *CER1* showed a consistent positive phenotype. These results indirectly reflect the fact that yield traits and stress resistance traits are often antagonistic to each other [70]. Functional exploration of these candidate genes has not been reported in *B. napus*. More studies are needed to reveal the roles of these genes.

In conclusion, our results revealed significant natural variations in germination traits under salt stress and yield-related traits. A total of 77 and 497 significant SNPs associated with three salt tolerance traits and six yield-related traits were detected, respectively. Based on GO analysis, we identified 19 candidate genes homologous to *Arabidopsis* genes known to be associated with salt stress and 7 promising candidates controlling both salt tolerance and yield traits. These SNP loci and candidate genes will be useful for future high salt resistance and high-yield breeding programs in rapeseed.

## 4. Materials and Methods

### 4.1. Plant Materials and Growth Conditions

A total of 146 *B. napus* accessions used in this study were provided by the Rapeseed Research Laboratory of the College of Agriculture, Northwest Agriculture and Forestry University. These germplasms were collected from China and were winter types. All accessions were grown in a greenhouse (light/dark 16/8 h photoperiod, 25/20 °C day/night temperature) for the subsequent salt-related trait evaluation. For the yield-related traits, 146 accessions were planted in the experimental field of Northwest Agriculture and Forestry University in Yangling, Shaanxi, China, in the winters of 2017, 2018, and 2019, and harvested in the springs of 2018, 2019, and 2020, respectively. The field trials in the three years were treated as three independent environments. Field trials followed a completely randomized design with no replicates in 2017, 2018, and 2019. Each accession was grown in a row with 10-12 plants, with a distance of 20 cm between plants within each row and 30 cm between rows. At the seedling stage, leaf tissues of all accessions were collected and frozen in liquid nitrogen for SLAF-seq analysis.

### 4.2. Phenotypic Evaluation and Statistical Analysis

For salt-related traits, seeds of uniform size were selected from 146 accessions and sterilized in 75% ethanol. Then, the seeds were washed with distilled water several times, and placed on glass petri dishes (90 mm) with double-layered filter paper for germination under control conditions and 230 mM NaCl conditions. Fifty seeds of each accession were measured per replicate and two replicates were assessed. The total number of the germinated seeds were counted after the 3rd and 7th day. Germination vigor was calculated as GV (%) = total number of germinated seeds on the 3rd day/total number of seeds × 100. Germination rate was calculated as GR (%) = total number of germinated seeds on the 7th day/total number of seeds × 100. Relative salt damage index was calculated as RSDI (%) = (control germination rate-treatment germination rate)/control germination rate × 100. For the yield-related traits, five plants with consistent growth per accession were selected for phenotypic determination during the maturity period, including silique length (SL), siliques on main inflorescence (SMI), seeds per silique (SS), thousand-seed weight (TSW), seed yield per plant (SYP), and dry weight per plant (DWP). The yield-related traits were determined as described previously [71]. Descriptive statistics were calculated using the mean values of all phenotypic data per accession and were evaluated using IBM SPSS version 20.0 (SPSS, Inc., Chicago, IL, USA). Pearson’s correlations between traits were calculated by IBM SPSS version 20.0 using the mean values of all traits across three years. Association analyses were carried out using the values per replicate for salt-related traits and values each year for yield-related traits. The frequency distribution of each trait was obtained using GraphPad Prism@ 8 (Version 8.0, GraphPad Software, Inc., San Diego, CA, USA). The correlation heatmap was obtained using the ChiPlot (www.chiplot.online/correlation_heatmap.html (accessed on 18 October 2022)).

### 4.3. SLAF Library Construction and Sequencing

The DNA of the 146 *B. napus* accessions was extracted from young leaf samples according to a modified cetyltrimethylammonium bromide (CTAB) method [72]. The concentration and quality of the DNA were determined using a Nanodrop 2000 spectrophotometer (Thermo Scientific, Waltham, MA, USA). The ‘Darmor-*bzh*’ reference *Brassica*_*napus*_v4.1 genome was used to design marker discovery experiments (http://www.genoscope.cns.fr/brassicanapus/data/, accessed on 18 October 2022). The SLAF library was constructed, as previously described with minor modifications [73]. HaeIII and Hpy166II were used to digest the genomic DNA. Fragments ranging from 464 to 514 base pairs (with indexes and adaptors) in size were excised and purified. Gel-purified products were then diluted for pair-end sequencing (each end 125 bp) on an Illumina High-Seq 2500 system (Illumina, Inc.; San Diego, CA, USA). Raw SLAF-seq data were analyzed using the Dual-Index software [74]. After filtering out the adapter reads, the sequencing quality was evaluated by estimating the guanine-cytosine (GC) content and Q30 ratio (Q value of 30 represents a 0.1% risk of error and 99.9% confidence). Thereafter, pair-end reads were clustered based on sequence similarity by BLAT. Polymorphic SLAF tags showed sequence polymorphisms across the accessions.

### 4.4. SNP Genotyping Analysis

The high-quality SLAF tags which show polymorphisms between accessions were aligned to the *B. napus* reference genome using Burrow’s Wheeler Alignment Tool (BWA) software [75]. SNPs were identified using the Genome Analysis Toolkit (GATK) and SAMtools [76,77]. We designated the intersection of SNPs obtained by both GATK and SAMtools as the reliable SNPs. Ultimately, SNPs with an integrity ratio of <0.8 and minor allele frequency (MAF) of <0.05 were filtered out via Plink2 software [78]. The sequencing data have been deposited in the sequence read archive under the accession number PRJNA883932.

### 4.5. Population Structure and Linkage Disequilibrium Analysis

To obtain accurate results for population structure analysis, we performed linkage disequilibrium (LD) pruning for SNPs derived from the filtered SNP. SNPs with pair-wise R^2^ > 0.2 were removed with a window size of 50 SNPs and window step of one SNP. Population analysis, phylogeny analysis, and principal component analysis (PCA) were performed. The population structure was analyzed by Admixture software [79]. The phylogenetic tree was constructed using MEGA5 software based on the neighbor-joining method, using the Kimura 2-parameter model with bootstrap repeated 1000 times [80]. PCA was performed using EIGENSOFT software [81]. The relative kinship between individuals was estimated using SPAGeDi software [82]. Linkage disequilibrium between SNPs was estimated using Plink2 software [78].

### 4.6. Genome-Wide Association Analysis

GWAS analysis was carried out using five different statistical GWAS models: the general linear model (GLM), mixed linear model (MLM), and compressed linear mixed model (CMLM) of TASSEL 5.0 Software [83], factored spectrally transformed linear mixed model (FaST-LMM) software [84], and efficient mixed-model association expedited (EMMAX) software [85]. The appropriate model for each trait was chosen based on the distribution of the *p*-values in the tail area using quantile-quantile plot (Q-Q plot) [86]. Markers with adjusted -log_10_ (*p*) > 4 (threshold) was considered as a significant SNP-trait association. A region of ~1 Mb that contains a few adjacent SNPs was regarded as a QTL region. The Q-Q plot and Manhattan plot were drawn using the R package “qqman” [87].

### 4.7. Genomic Comparison of Present and Previous QTLs for All Traits

In order to further verify the reliability of the QTLs, QTL mapping studies co-localized with the present study were collected for salt tolerance [13,17,18,19,20,21,22,23,39] and yield-related traits including SL, SS, SMI, TSW, SYP, and DWP [10,11,12,36,48,88,89,90,91,92,93,94,95,96,97,98,99,100,101,102,103] conducted on *B. napus*. QTLs from previous studies and the present study were compared to the physical genomic regions of *B. napus*, “Darmor-*bzh*”. If only one flanking marker could be aligned to the reference genome, we used a uniform area of 1 cM to delimit all QTLs.

## Figures and Tables

**Figure 1 ijms-23-15892-f001:**
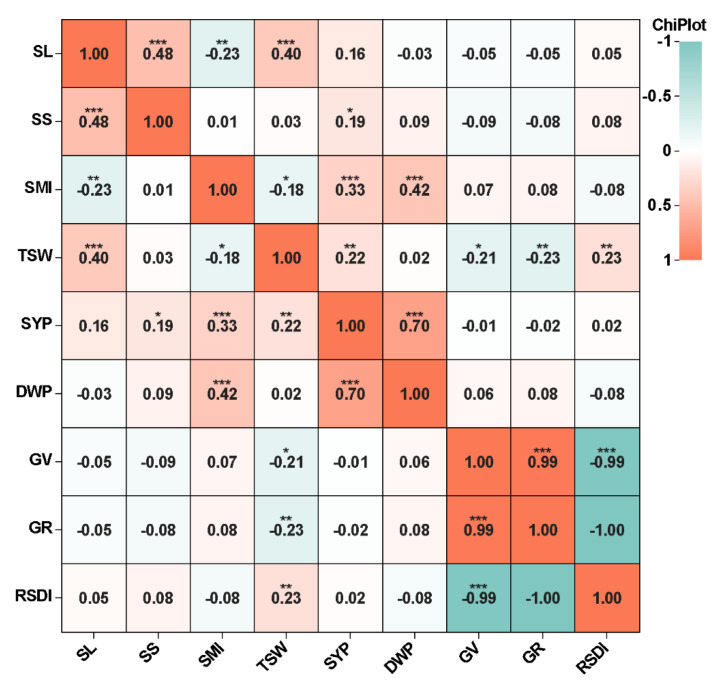
Correlation analysis of all traits for 146 rapeseed accessions. The numbers represent the value of the correlation coefficient. The color of the box indicates the degree of correlation between the row trait and the column trait. The orange box indicates a positive correlation between the two traits; the green box indicates a negative correlation between the two traits. SL, silique length; SMI, siliques on main inflorescence; SS, seeds per silique; TSW, thousand-seed weight; SYP, seed yield per plant; DWP, dry weight per plant; GV, germination vigor; GR, germination rate; RSDI, relative salt damage index. (*), (**), (***) significance levels of 0.05, 0.01 and 0.001, respectively.

**Figure 2 ijms-23-15892-f002:**
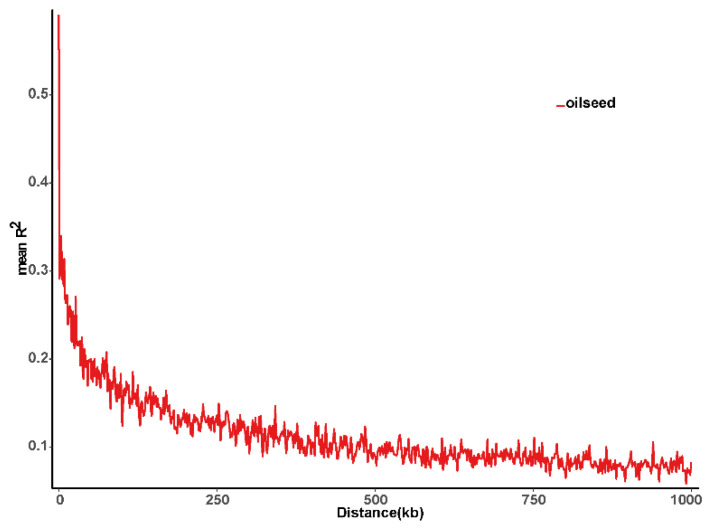
The linkage disequilibrium (LD) decay for the 146 rapeseed accessions.

**Figure 3 ijms-23-15892-f003:**
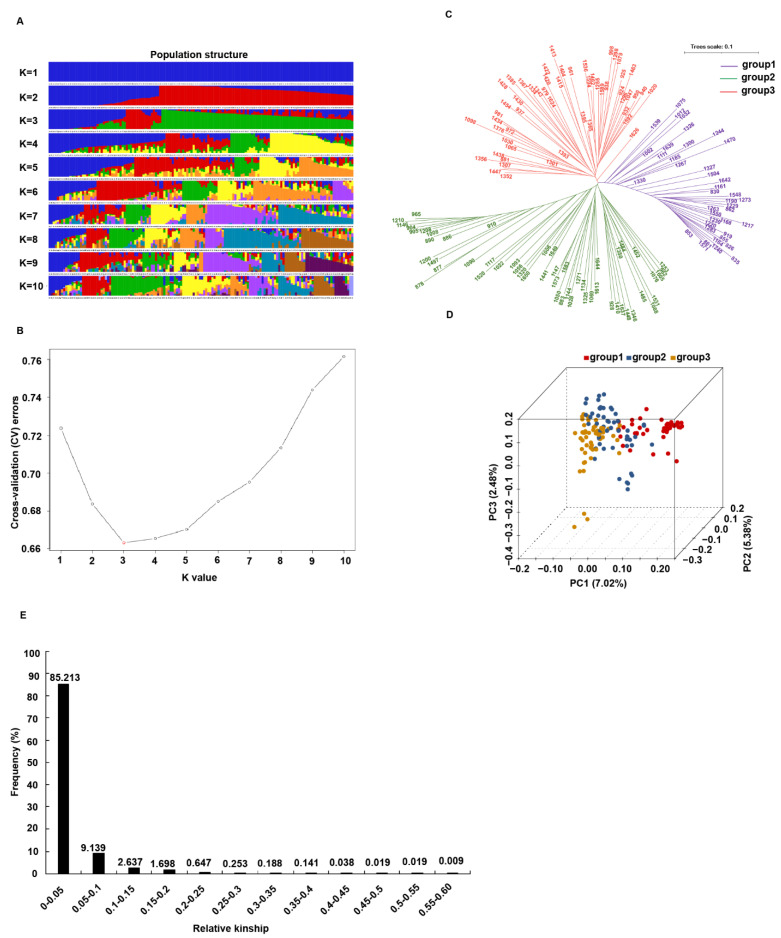
The population structure and relative kinship analysis for the 146 rapeseed accessions. (**A**) The population structure of rapeseed accessions estimated by ADMIXTURE. The colors represent separate groups. (**B**) Cross validation error rate for each K-value. (**C**) Phylogenetic tree of 146 rapeseed accessions. Different colors represent different subgroups: group1-group3. (**D**) Three-dimensional PCA scatter plots. The accessions were divided into three subgroups: group1–group3. The three axes refer to PC1, PC2, and PC3, respectively. Each dot represents a sample. (**E**) Distribution of pairwise relative kinship of rapeseed accessions.

**Figure 4 ijms-23-15892-f004:**
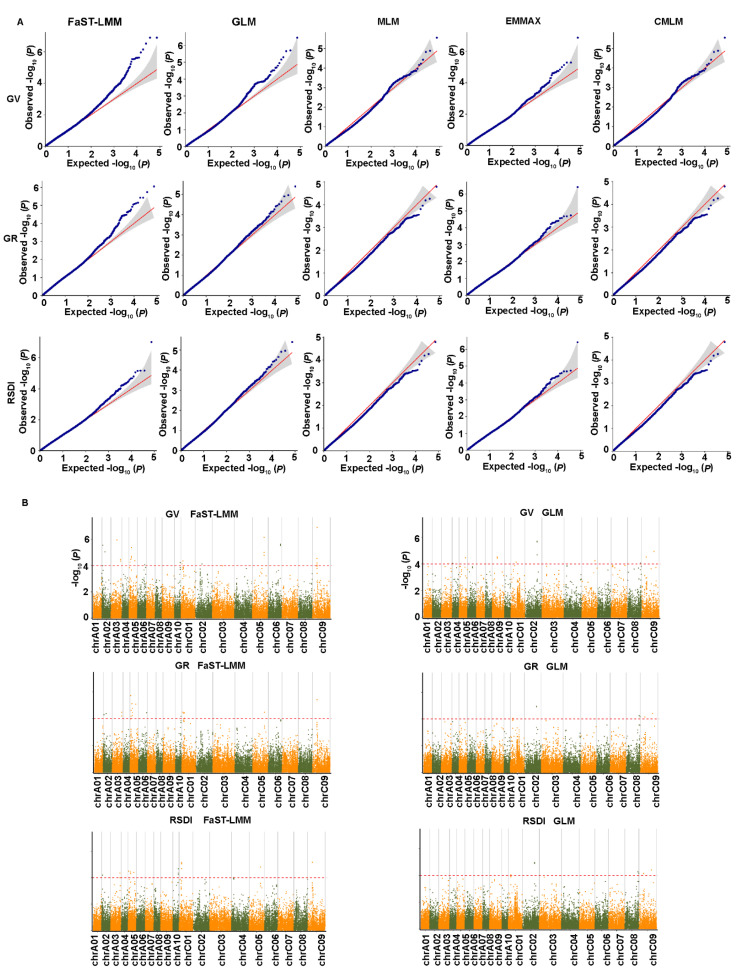
Quantile-quantile and Manhattan plots from association analysis of salt tolerance traits. (**A**) Quantile-quantile plots using the five statistical models. The red line represents the 45° centerline, and the gray area is the 95 % confidence interval of the scattered points. (**B**) Manhattan plots using the optimal statistical models. The X-axis indicates the nineteen chromosomes and the Y-axis represents −log10 (*p*) values of the SNP marker. The red dashed line represents the significance threshold (*p* < 0.0001). GLM, general linear model; MLM, mixed linear model; FaST-LMM, factored spectrally transformed linear mixed model; EMMAX, efficient mixed-model association expedited; CMLM, compressed linear mixed model. GV, germination vigor; GR, germination rate; RSDI, relative salt damage index. The graphs show the results of replicate 1.

**Figure 5 ijms-23-15892-f005:**
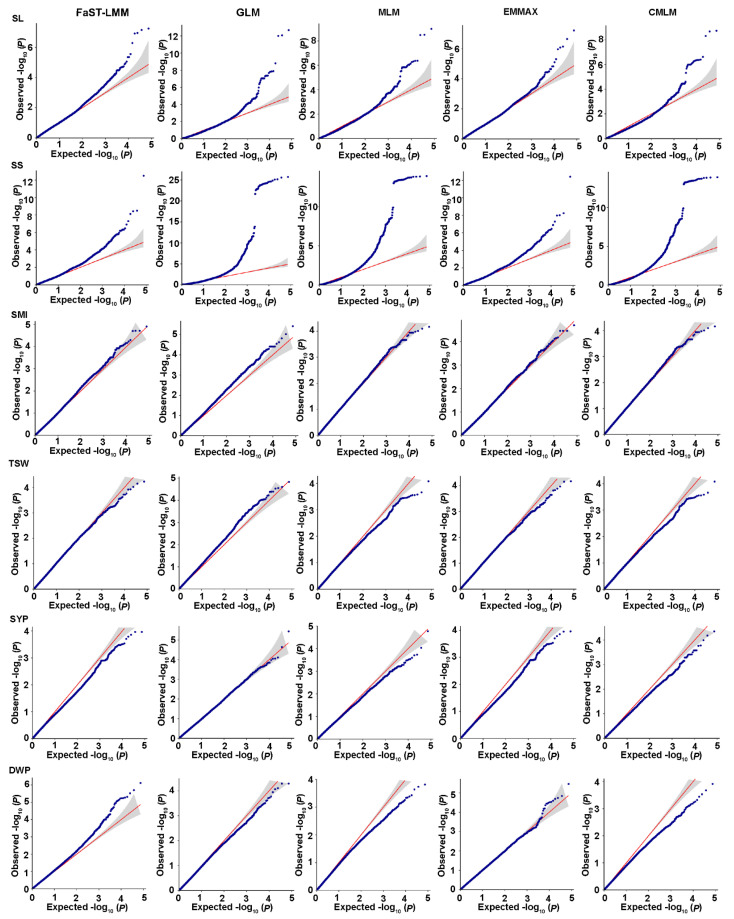
Quantile-quantile plots from association analysis of six yield-related traits. The red line represents the 45° centerline, and the gray area is the 95% confidence interval of the scattered points. GLM, general linear model; MLM, mixed linear model; FaST-LMM, factored spectrally transformed linear mixed model; EMMAX, efficient mixed-model association expedited; CMLM, compressed linear mixed model; SL, silique length; SMI, siliques on main inflorescence; SS, seeds per silique; TSW, thousand-seed weight; SYP, seed yield per plant; DWP, dry weight per plant. The graphs show the results in a year (2018).

**Figure 6 ijms-23-15892-f006:**
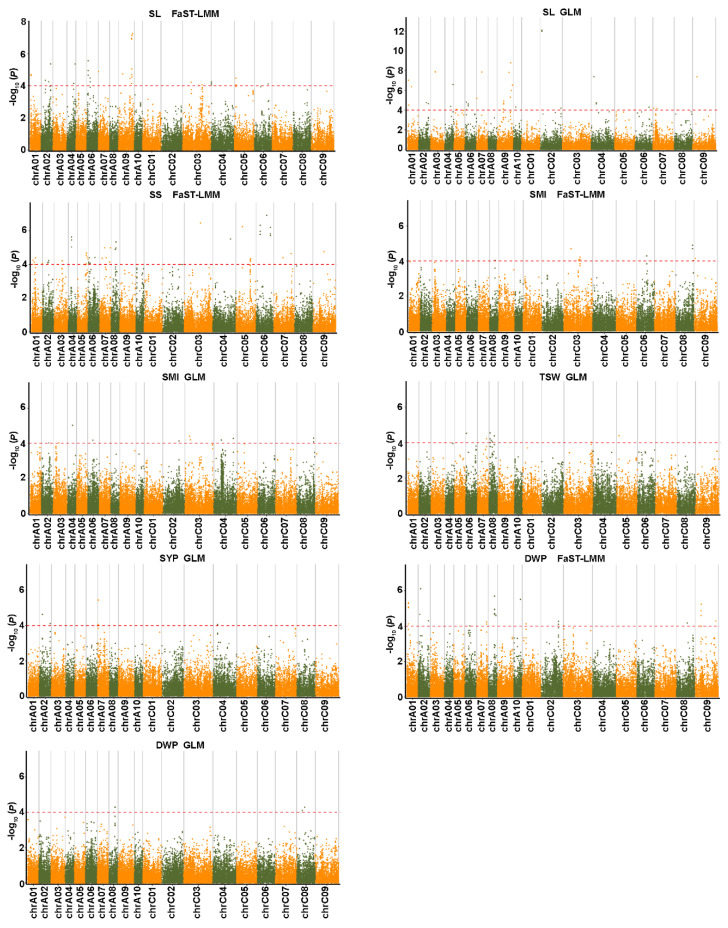
Manhattan plots from association analysis of six yield-related traits using the optimal statistical models. The X-axis indicates the nineteen chromosomes and the Y-axis represents −log10 (*p*) values of the SNP marker. The red dashed line represents the significance threshold (*p* < 0.0001). GLM, general linear model; FaST-LMM, factored spectrally transformed linear mixed model; SL, silique length; SMI, siliques on main inflorescence; SS, seeds per silique; TSW, thousand-seed weight; SYP, seed yield per plant; DWP, dry weight per plant. The graphs show the results in a year (2018).

**Table 1 ijms-23-15892-t001:** Phenotypic variation of 146 rapeseed accessions.

Trait	Year	Mean ± SD	Range	Variance	CV (%)
SL (cm)	2018	5.74 ± 0.91	4.32–10.86	0.83	15.85
	2019	6.56 ± 1.13	4.34–12.71	1.28	17.23
	2020	7.72 ± 1.06	5.19–12.06	1.12	13.73
SS (number)	2018	15.76 ± 2.60	9.00–22.93	6.75	16.50
	2019	22.84 ± 3.90	9.94–34.73	15.19	17.08
	2020	26.50 ± 4.05	12.67–35.10	16.41	15.28
SMI (number)	2018	50.02 ± 11.69	28.40–88.27	136.58	23.37
	2019	61.02 ± 12.30	26.33–111.67	151.27	20.16
	2020	71.01 ± 14.23	25.25–111.67	202.44	20.04
TSW (g)	2018	5.07 ± 0.86	3.07–7.28	0.75	16.96
	2019	4.02 ± 0.83	2.20–7.27	0.70	20.65
	2020	3.11 ± 0.72	1.69–5.45	0.52	23.15
SYP (g)	2018	16.20 ± 5.03	4.10–33.24	25.26	31.05
	2019	26.12 ± 9.77	6.67–50.00	95.45	37.40
	2020	22.42 ± 8.53	5.37–47.00	72.80	38.05
DWP (g)	2018	77.60 ± 15.90	38.67–127.33	252.69	20.49
	2019	99.48 ± 27.33	46.67–183.33	747.01	27.47
	2020	144.31 ± 28.69	80.00–215.00	823.28	19.88
GV (%)		11.14 ± 14.86	0.00–72.00	220.90	133.00
GR (%)		14.13 ± 17.00	0.00–76.50	288.89	120.00
RSDI (%)		85.87 ± 17.00	23.50–100.00	288.89	20.00

SL, silique length; SMI, siliques on main inflorescence; SS, seeds per silique; TSW, thousand-seed weight; SYP, seed yield per plant; DWP, dry weight per plant; GV, germination vigor; GR, germination rate; RSDI, relative salt damage index; SD, standard deviation; CV, coefficient of variation.

**Table 2 ijms-23-15892-t002:** Co-localized SNPs’ association between the salt tolerance and the yield-related traits.

Yield-Related Traits					Salt-Related Traits					
Chr	Trait	QTL	Interval (bp)	Peak SNP (bp)	Trait	Chr	SNP	SNP Position (bp)	Rep1	Rep2
A02	DWP/SMI	*pqA2.1*	1,897,176–3,836,751	2,397,176/3,336,751	GV/GR/RSDI	A02	rsA02_2496879	2,496,879	√	√
					GR	A02	rsA02_3530820	3,530,820		√
	SL	*qSL18–A2.1*	7,907,803–8,907,803	8,407,803	GV/GR	A02	rsA02_8228124	8,228,124	√	√
		*qSL19–A2.2*	8,051,356–9,051,356	8,551,356						
A05	SL	*qSL18–A5.2*	6,194,267–7,194,267	6,694,267	GV/GR/RSDI	A05	rsA05_6670181	6,670,181	√	√
					GV/GR/RSDI	A05	rsA05_6692257	6,692,257	√	√
					GV	A05	rsA05_6692275	6,692,275	√	√
	SMI	*qSMI20–A5.1*	18,168,998–19,168,998	18,668,998	GR	A05	rsA05_18700305	18,700,305		√
A09	SL	*qSL18–A9.6*	31,410,855–32,410,855	31,910,855	GR	A09	rsA09_31840807	31,840,807		√
C04	SL	*qSL18–C4.2*	5,051,524–6,051,524	5,551,524	RSDI	C04	rsC04_5933538	5,933,538		√
					RSDI	C04	rsC04_5933539	5,933,539		√
					RSDI	C04	rsC04_5933579	5,933,579		√
					RSDI	C04	rsC04_5933820	5,933,820		√
C09	DWP/SS	*pqC9.1*	12,405,354–13,405,354	12,521,248/12,521,335/12,724,461/12,976,490/12,976,522/12,905,354	GR	C09	rsC09_12521248	12,521,248		√
					GV/GR	C09	rsC09_12521335	12,521,335	√	√
					GV	C09	rsC09_12645583	12,645,583	√	
					GV	C09	rsC09_12645586	12,645,586	√	
					GV/GR/RSDI	C09	rsC09_12949548	12,949,548	√	√
					GV/GR/RSDI	C09	rsC09_12949561	12,949,561	√	√

Rep1/2, two biological replicates; Chr, Chromosome; SL, silique length; SMI, siliques on main inflorescence; SS, seeds per silique; DWP, dry weight per plant; GV, germination vigor; GR, germination rate; RSDI, relative salt damage index.

**Table 3 ijms-23-15892-t003:** The candidate genes controlling both salt tolerance and yield.

Gene Name	Chr	Site_strat (bp)	Site_end (bp)	Distance (bp)	SNP Location (bp)	Orthologous Gene ID in *Arabidopsis*	Gene Name in *Arabidopsis*	Function Description in *Arabidopsis*
*BnaA06g33990D*	A06	22,513,737	22,516,788	5′_42149	22,471,588, 22,471,644	*AT2G02820*	*MYB88*	a putative transcription factor (MYB88)
*BnaA03g33890D*	A03	16,409,029	16,410,113	5′_33265	16,375,764	*AT3G15500*	*ANAC55*	NAC domain-containing protein 55
*BnaC06g34470D*	C06	33,929,460	33,934,599	3′_86296	33,843,164, 33,843,483, 33,843,558	*AT1G73660*	*M3KDELTA6*	serine/threonine–protein kinase EDR1
*BnaA02g14490D*	A02	8,200,831	8,202,439	3′_25685	8,228,124	*AT1G69700*	*HVA22C*	HVA22–like protein c
*BnaA03g50340D*	A03	26,148,221	26,149,401	5′_23636	26,173,037	*AT2G18960*	*OST2*	a plasma membrane proton ATPase
*BnaA03g50610D*	A03	26,258,416	26,258,990	5′_85379	26,173,037	*AT4G33730*	*ATCAPE1*	CAP-derived peptide1
*BnaA02g05520D*	A02	2,548,591	2,548,857	3′_51712	2,496,879	*AT5G22270*	*SIED1*	salt-induced and EIN3/EIL1-dependent 1
*BnaA02g05590D*	A02	2,569,927	2,571,416	3′_73048	2,496,879	*AT5G22360*	*VAMP714*	Vesicle-associated membrane protein 714
*BnaA05g11750D*	A05	6,644,918	6,646,115	3′_46142	6,670,181, 6,692,257, 6,692,275	*AT4G18780*	*LEW2*	cellulose synthase A catalytic subunit 8
*BnaA05g11880D*	A05	6,747,961	6,752,424	3′_55704	6,670,181, 6,692,257, 6,692,275	*AT2G30580*	*DRIP2*	C3HC4 RING-domain-containing ubiquitin E3 ligase
*BnaA05g20580D*	A05	15,949,088	15,952,214	3′_99304	16,051,518	*AT3G19490*	*NHD1*	member of Na +/H+ antiporter-putative family
*BnaA05g20640D*	A05	15,986,304	15,989,708	5′_61810	16,051,518	*AT3G19420*	*PEN2*	phosphatase with low tyrosine phosphatase activity
*BnaA10g22560D*	A10	15,173,521	15,178,533	5′_10326	15,188,859	*AT5G09410*	*CAMTA1*	Calmodulin-binding transcription activator 1
*BnaA10g22820D*	A10	15,266,296	15,267,019	3′_77437	15,188,859	*AT5G08620*	*STRS2*	DEAD2013box ATP-dependent RNA helicase 25
*BnaA10g22850D*	A10	15,278,666	15,281,149	3′_89807	15,188,859	*AT5G08590*	*SNRK2.1*	serine/threonine-protein kinase SRK2G
*BnaA10g22880D*	A10	15,287,668	15,290,955	5′_98809	15,188,859	*AT5G08560*	*WDR26*	a WD-40-repeat-containing protein
*BnaC02g31110D*	C02	33,207,103	33,208,273	5′_11078	33,196,025, 33,196,054	*AT5G44650*	*CEST*	chloroplast protein-enhancing stress tolerance
*BnaC09g15950D*	C09	12,945,604	12,947,996	5′_1552	12,949,548, 12,949,561	*AT1G56570*	*PGN*	putative pentatricopeptide repeat-containing protein
*BnaC09g16050D*	C09	13,037,634	13,039,703	5′_88086	12,949,548, 12,949,561	*AT1G02205*	*CER1*	protein ECERIFERUM 1

Chr, Chromosome; Distance (bp), the distance between significant SNPs and candidate genes; GV, germination vigor; GR, germination rate; RSDI, relative salt damage index; GLM, general linear model; FaST-LMM, factored spectrally transformed linear mixed model.

## Data Availability

The datasets generated during the current study are available in the Sequence Read Archive (SRA) database of National Center for Biotechnology Information (NCBI) under accession PRJNA883932 (https://dataview.ncbi.nlm.nih.gov/object/PRJNA883932 (accessed on 10 October 2022)).

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
