# Peer review of "Genome-Wide Association Studies of Salt Tolerance at the Seed Germination Stage and Yield-Related Traits in Brassica napus L."

_ijms, 2022, doi:10.3390/ijms232415892_

Round 1

Reviewer 1 Report

Authors are advised to go through the manuscript for minor scientific editing. The suggestions and comments are attached herewith.

Reviewer 2 Report

The manuscript "Genome-wide association studies of salt tolerance at the seed germination stage and yield-related traits in Brassica napus L. " by Zhang et al. is a comprehensive and well-executed study regarding the correlation of salt stress responses in seed germination and yield with the occurrence of SNPs. The methodology is well described and is easy to follow. The data and the statistical analysis were presented in detail. The discussion is methodical. 

I see no fault in this manuscript and recommend publication.

Reviewer 3 Report

The paper written by Zhang et al. is a coherent analysis of genome-wide studies with view on germination, salt tolerance and yield. They look  for traits in Brassica napus, which will be useful for 409 future high salt resistance and high yield breeding programs in rapeseed. The paper is well written, and documented. The statistical analysis seems to be properly verified. There are just minor editorial issues, which could be addressed to this paper.

Some genes names are not italics, as f.eg. in Lines: 363, 364, 368, 393. Please check it carefully throughout the text

It is good to write the full name before abbreviation at first use. There is lack of full name for lots of abbreviations in 2.1 section. The full names are under Table 1.

When you listed species names as in line 62 it is good to use just English or Latin names, but not to mix them.
